# Morphological Adaptation in the Jejunal Mucosa after Iso-Caloric High-Fat versus High-Carbohydrate Diets in Healthy Volunteers: Data from a Randomized Crossover Study

**DOI:** 10.3390/nu14194123

**Published:** 2022-10-04

**Authors:** Anna Casselbrant, Ville Wallenius, Erik Elebring, Hanns-Ulrich Marschall, Bengt R. Johansson, Herbert F. Helander, Lars Fändriks

**Affiliations:** 1Institute of Clinical Sciences, Sahlgrenska Academy, University of Gothenburg, SE-41345 Gothenburg, Sweden; 2Department of Surgery, Region Västra Götaland, Sahlgrenska University Hospital, SE-41345 Gothenburg, Sweden; 3Institute of Medicine, Sahlgrenska Academy, University of Gothenburg, SE-41345 Gothenburg, Sweden; 4Institute of Biomedicine, Sahlgrenska Academy, University of Gothenburg, SE-41345 Gothenburg, Sweden

**Keywords:** jejunum, mucosa, microvilli, mitochondria, high-fat, high-carbohydrate, diet

## Abstract

Background and aims: The conditions for jejunal glucose absorption in healthy subjects have not been thoroughly studied. In this study we investigated differences in *the jejunal villi* enlargement factor, as well as *ultrastructural aspects* of the surface enterocytes and mitochondria, comparing 2 weeks of high-carbohydrate (HCD) versus high-fat diets (HFD). We also measured the ketogenesis rate-limiting enzyme 3-hydroxy-3-methylglutaryl-CoA synthase (HMGCS2) in relation to jejunal mitochondria. Methods: A single-centre, randomized, unblinded crossover study in 15 healthy volunteers ingesting strictly controlled equicaloric diets (either HCD or HFD), with 60% energy from the respective source. An enteroscopy was carried out after 2 weeks of each diet and jejunal mucosal biopsies were acquired. Conventional histology, immunofluorescent staining, transmission electron microscopy and confocal microscopy were used. Results: The villi did not demonstrate any change in the epithelial enlargement factor. Despite an increased mitosis, there were no changes in apoptotic indices. However, the ultrastructural analysis demonstrated a significant increase in the enlargement factor at the bases of the villi. The mitochondria demonstrated increased amounts of cristae after the HFD. The confocal microscopy revealed increased HMGCS2 per mitochondrial marker at the top of the villi after the HFD compared to the HCD. Conclusion: There is a morphometric adaption in the jejunal mucosa following the 2-week diets, not only on a histological level, but rather on the ultrastructural level. This study supports the notion that mitochondrial HMGCS2 is regulated by the fat content of the diet and is involved in the expression of monosaccharide transporters.

## 1. Introduction

The human small intestinal mucosa is a large and wide organ and has a complex geometry with crypts, villi and microvilli. The total area of the luminally faced gastrointestinal epithelial membrane is between 30 and 40 m^2^, of which 90–95% belongs to the small intestine [1]. Another remarkable feature of the small intestinal mucosa is that the cell turnover rate is among the highest of all of the body’s tissues [2]. The vast majority of cells, *enterocytes*, move from the crypts towards the tips of the villi during continuous differentiation and will eventually enter apoptosis, and are then extruded into the lumen [3]. The migration time from the dividing stem cells in the crypts to the tips of villi for enterocytes is between 3 to 5 days [4]. This implies that a complete renewal of the human small intestinal surface epithelium occurs every few days. The small intestinal mucosa is exposed to extremely variable environmental factors (for example, toxins and microbiota from food) and with endogenous aggressive factors (for example, proteolytic enzymes and bile salts) [5]. For the small intestinal mucosa it is, therefore, necessary to be able to rapidly change phenotype depending on the luminal contents. The first part of the small intestine, *the duodenum*, is rather short, approximately 0.3 m. It is characterized by a mixture of the gastric emptying of food with pancreatic secretions and bile. The present investigation studied the next part of the small intestine, the *jejunum*, which represents the largest part of the small intestine. The jejunum arbitrarily enters into the last part of the small intestine, the *ileum*. This part has a higher number of microbiota and is mostly known for its uptake of biliary components. The ileum prepares the chyme for the large reservoir of microbiota in the large intestine, the colon. We chose to study the *proximal jejunum* because it has the highest digestion and absorption of nutrients and minimal microbiota. 

Consumption of various diets, particularly a high-fat diet (HFD), has been identified to regulate key aspects of whole-body metabolism [6]. In this study, we therefore wanted to study how the jejunal mucosa morphologically adapts after an extreme diet in healthy volunteers. Human data in the literature are often derived from the non-pathological parts of the mucosae resected in conjunction with a suspected or verified medical condition, for example tumors or inflammatory processes [1]. In the present study, we tried to avoid these possible confounding factors by studying the jejunum from healthy human volunteers. The participants ingested two individualized iso-caloric diets, either a high-carbohydrate diet (HCD) or a high-fat diet (HFD) (each consisting of 60 percent of the total energy content (60%E) of either macrocomponent), over two weeks and in a random-crossover fashion. Intra-individual comparisons of the HCD- and the HFD-exposed mucosae were carried out at the end of the dietary period. Two reports have already been published regarding this study: a first report on the systemic glucose clearance following a mixed meal test performed on day 12 for each diet [7], and a second on glucose transport in jejunal biopsies taken on day 14 for each diet [8]. The present third paper is a morphological report with the aim to describe the *epithelial enlargement of the jejunal villi* comparing the HCD and HFD, including immunohistochemical aspects of the proliferation and apoptosis. We also studied the ultrastructural aspects of the surface enterocytes and their mitochondria. In the previous study, we concluded that jejunal glucose absorption is decreased by the HFD via the ketogenesis rate-limiting enzyme mitochondrial 3-hydroxy-3-methylglutaryl-CoA synthase (HMGCS2) [8]. Therefore, as a final aim, we wanted to study the co-location of HMGCS2 and mitochondria following each diet.

## 2. Material and Methods

### 2.1. The Clinical Study

The present study was performed in accordance with the Declaration of Helsinki and was approved by the Regional Ethical Review Board in Gothenburg, Sweden (ethics application number 807–11). The study was registered at ClinicalTrials.gov (NCT02088853) and was performed at the Dept. of Surgery, Sahlgrenska University Hospital, Gothenburg, between February and December 2014. All study participants provided written informed consent. Further details on the design of the study and other outcomes have been published previously [7] (see also the CONSORT figure in ref [7]). Briefly, the sample size was calculated using the primary outcome (mucosal enlargement factor); the secondary outcomes have been reported previously (Ussing-chamber based). The inclusion criteria were: voluntary participation, self-reported good general health status, age between 18 and 65 years and body mass index (BMI) between 18 and 25 kg/m^2^. Exclusion criteria were: overweight or obesity (BMI > 25 kg/m^2^); history of drug abuse or smoking; use of prescription medications within the previous 14 days (with the exception of contraceptives); pregnancy or breast feeding, or women at childbearing age not using adequate birth control. The study was designed as a single-centre, randomized, unblinded crossover study in healthy volunteers ingesting strictly controlled equicaloric diets either rich in carbohydrates or rich in fat (with predominantly saturated fat), with 60% energy from the respective source (see the dietary composition at: https://www.mdpi.com/article/10.3390/nu13103322/s1 (accessed on 30 August 2022)). The diets were adjusted to the participant’s body-weight according to the Mifflin St Jeor equation [9]. The order of the study diets was randomized in blocks due to cooking logistics. A study nurse randomly assigned the patients to start the study in one of the study diet blocks. The diets were administered over two 14-day periods in random order with an intervening wash-out period of a minimum of two weeks. The participants were instructed not to eat anything but the food and drinks provided by the laboratory, but were free to drink extra tap water if needed. On day 12 of each diet period, the participants were subject to a mixed meal test, as reported elsewhere [7], and on day 14 of each diet period the participants returned for enteroscopy after an overnight fast.

### 2.2. Enteroscopy and Specimen Preparation

After sedation with midazolam and alfentanil, a thin calibrated enteroscope was introduced into the gastroduodenum and proximal jejunum. A total of 5 to 10 biopsies were obtained from the jejunum approximately 50 cm distal to the ligament of Treitz. Four to six jejunal biopsies were either snap frozen in liquid nitrogen or chemically fixated for later analyses. The chemically fixated biopsies were fixed in phosphate-buffered 4% formaldehyde, dehydrated, embedded in paraffin and cut in 5-μm-thick sections (see Section 2.3); in a subsample, the embedded biopsies were cut in 15-μm-thick sections (see Section 2.5). Biopsies from the six first participants were prepared for electron-microscopy (see Section 2.4). We tried to perform paired comparisons (HCD vs. HFD), but this was not always possible due to lack of material or technical disturbances in the biopsy material from the participants. The number of successful pairwise comparisons is given in the figures.

### 2.3. Morphometry and Cell Proliferation

Morphological features of the mucosal biopsies were evaluated in a coded fashion (Leica Microsystem, Wetzlar, Germany). The length of villi and crypts, respectively, were assessed after staining with haematoxylin and eosin and in sections that allowed optimal orientation. The villus-derived amplification factor (the mucosal surface in relation to the relatively flat muscularis mucosae) was assessed by the gridline intersection method [10].

For immunostaining, sections were rehydrated; antigens were retrieved by boiling for 1 min in 50 mM borate buffer (pH 8.0), blocked in 5% normal goat serum and incubated in primary antibody against Rb-a-Ki-67 (1:100, Biocare Medical, Concord, CA, USA) for 1 h at room temperature. Immunoreactivity was visualized by the diaminobenzidine (DAB) method, and sections were contrasted for 30 s in Mayer’s haematoxylin. Mitotic activity by use of Ki-67 immunostaining was expressed as % stained crypt cells, and was used as marker of cell proliferation. Biomarkers of apoptosis were analysed with immunoblotting. Frozen samples were sonicated in ice-cold protein extraction buffer (10 mM potassium phosphate buffer, pH 6.8, containing 1 mM ethylenediaminetetraacetic acid, 10 mM 3-[(3-cholamidopropyl) dimethylammonio]-1-propanesulfonate) and Complete™ protease inhibitor cocktail. After sonication, each homogenate was centrifuged (10,000× *g*, 10 min, 4 °C) and the protein concentration of the supernatant was quantified with the standard Bradford method. Total protein samples were diluted in SDS buffer and heated at 70 °C for 10 min before being loaded on NuPage 10% Bis-Tris gel and electrophoresis was run using MOPS buffer. After the electrophoresis, proteins were transferred to polyvinyldifluoride membranes using the iBlot blotting system. Membranes were incubated with primary antibodies against Caspase 3 (1:500, HPA002643, Atlas Antibodies AB, AlbaNova University Center, Stockholm, Sweden), BAX (1:2000, ab182733, Abcam, Cambridge, UK), Bcl-2 (1:500, ab196495, Abcam), and Bcl-XL (1:1000, MA5-15142, ThermoFisher Scientific, Rockford, IL, USA) followed by rabbit HRP-linked secondary antibody (1:1000, #7074, Cell Signaling Technology, Leiden, The Netherlands). Chemiluminescense was developed with addition of WesternBright Quantum reagents (K-12042, Advansta Corporation, Menlo Park, CL, USA). Images were captured by a Chemidoc™ XRS system, and semi-quantification was performed using Quantity One software (Version 4.0.3, Mac, BioRad Laboratories, Hercules, CA, USA). Membranes were sequentially stripped using ReBlot Plus Mild solution (Millipore, Temecula, CL, USA). Glyceraldehyde 3-phosphate dehydrogenase (GAPDH, 1:1000, IMG-5143A, Imgenex, San Diego, CL, USA) was used as control for equal loading for each tested sample. Data are presented as the ratio between optical density of primary antibody and of GAPDH in each sample.

### 2.4. Transmission Electron Microscopy (TEM)

The biopsies were kept in ice-cold Ringer solution for ~30 min before fixation in ice-cold Karnovsky fixative (2% formaldehyde, 2.5% glutaraldehyde, 0.02% Na-azide in 0.05 M sodium cacodylate, pH 7.2). After fixation overnight, the specimens were post-fixed in 1% OsO_4_, and 1% K_4_Fe(CN)_6_ in 0.05 M sodium cacodylate, for 2 h at 4 °C. A bloc staining followed in 0.5% uranyl acetate in distilled water for 1 h at room temperature in darkness. The tissues were then rinsed in distilled water, dehydrated in rising concentrations of ethanol, followed by 100% acetone, and were infiltrated and embedded in Agar 100 resin. Sixty nm ultramicrotome sections were photographed in an electron microscope (Zeiss Leo 912 Omega, Oberkochen, Germany) at primary magnifications ×4000, ×8000, ×25,000 or ×31,500. To standardize the measurements, the orientation of the sections must meet certain criteria: straight microvilli bordering the lumen; clear cell nucleus in the middle; and infranuclear cytoplasm sitting on the basement membrane. We counted the number of intersections between a square grid (square size 12 × 12 mm^2^) and the cytoplasm, which gives an estimate of the size of the cytoplasm [10]. Length and diameter of the microvilli, as well as surface amplification due to microvilli, were measured in ten contiguous cells with a random start. Gridline intersections were counted only when the plasma membrane was clearly visible, and were related to the number of intersections of a reference line drawn in parallel with the base of the microvilli. In addition, we determined the number of mitochondria, the surface density of the mitochondrial cristae and the length of the four longest mitochondria. In each individual, up to 6 cells were measured at each level and each diet. All sections were coded until after all measurements had been concluded. The higher magnifications permitted us to estimate the amounts of cristae inside the mitochondria using the grid intersection method, then focusing on the supranuclear and the infra-nuclear parts of enterocytes. This was done for technical reasons, as it is impossible to photograph an entire cell at the same time.

### 2.5. Confocal Microscopy

Immunofluorescent double stainings were performed with the mitochondrial ketogenesis enzyme HMGCS2 and mitochondrial marker TOMM20. The sections were rehydrated, antigens were retrieved and sections were blocked in 5% normal goat serum (31872, ThermoFisher Scientific, Rockford, IL, USA) before incubation with primary antibodies HMGCS2 (1:150, sc-393256, Santa Cruz Biotechnology, Dallas, TX, USA) and TOMM20 (1:150, ab186734, Abcam, Cambridge, UK) overnight at 4 °C. Slides were then washed and incubated with the secondary antibodies Alexa Fluor 488 goat anti-mouse (1:1000, A11001, ThermoFisher Scientific) and Alexa Fluor 568 goat anti-rabbit (1:1000, A11011, ThermoFisher Scientific) for 2 h at room temperature and counter-stained with Hoechst staining (1:10,000 in PBS, H6024, Sigma-Aldrich, Saint Louis, MO, USA). Sections with blocking buffer instead of primary antibodies were included as negative controls. Confocal imaging was performed at the Centre for Cellular Imaging at Sahlgrenska Academy using an LSM 700 inverted laser scanning confocal microscope (Zeiss, Jena, Germany) with Zen 2012 software (black edition, Zeiss). All images were captured with the same settings using an apochromatic 20X/0.8 objective. For the longest wavelength, the pinhole was set to 1 airy unit and this pinhole size was applied to both channels. For each sample, 24 optical slices (0.998 μm thickness) were made in z-dimension around the center of the sample. For each sample, scanning images were acquired from the top, middle and base of the villus. Acquired images were quantified for intensity signal of each channel using Fiji version 2.1.0/1.53c [11]. Negative controls were used for background correction. For each acquired image, the integrated intensity was summed over the z-stack for each channel.

### 2.6. Statistical Analyses

Statistical outcomes of the paired Western blot, immunohistochemistry, etc. were analysed with the Wilcoxon signed-rank test. The 2-way ANOVA with Sidak’s multiple comparison test was used for confocal microscopy. The standard errors of the mean (SEM) are presented in figures. A *p*-value of ≤0.05 was considered significant. All analyses were performed using Prism 9 for Mac OS X (GraphPad Software Inc. San Diego, CA, USA).

## 3. Results

Fifteen healthy volunteers (seven females, eight males) with a mean body weight of 72 ± 3 kg ingested two weeks of each iso-caloric diet; beginning either with the HCD or the HFD, in a crossover randomized order [7]. One female participant had to be excluded after the first dietary period due to an unplanned pregnancy. The two study diets were generally well accepted and tolerated. The majority of the participants reported that the HCD was more challenging because of its larger volume, whereas the HFD was always easily ingested. On day 14 in each dietary period, the volunteers underwent enteroscopy with biopsy sampling from the jejunum. The jejunal epithelial monosaccharide transporters in these biopsies have been reported elsewhere [8].

### 3.1. Intraindividual Comparison between HCD and HFD Did Not Reveal any Changed Mucosal Enlargement Factor

Figure 1A is an example of a jejunal biopsy with the villus length (a), the crypt depth (b) and the muscularis mucosae (c) marked. The jejunal villus and crypt depths were not significantly different between the HCD and the HFD and the mucosal surface enlargement factor did not differ between the two diets (Figure 1B). The majority of the Ki-67 positive cells were found close to the bottom of the crypts of Lieberkühn, as shown in Figure 1A. The crypt mitotic activity, assessed as %Ki-67 activity, was significantly higher after the HCD compared to the HFD (Figure 1B, *p* = 0.027).

The general biomarker for apoptosis, Caspase 3, as well as the anti-apoptotic oncoprotein Bcl-2 did not differ significantly between diets. However, an increase in the pro-apoptotic protein BAX could be seen after the HCD as compared to after HFD (Figure 1C, *p* = 0.034). Despite this, the ratio between pro- and antiapoptotic proteins (as a measurement of apoptotic probability) was not significantly different between the two diets (Figure 1C).

### 3.2. Ultrastructural Assessment of the Enterocyte Surface Revealed a Larger Surface Enhancement following HCD at The Base of the Villus

Figure 2A shows light microscopy (×10) and Appendix A shows electron microscopy (4 × 4000) illustrates the *top* region and the *base* region that were analysed in electron microscopy presentations. Figure 2B,C show examples of successful pairwise (HCD and HFD) sample collections from the same individual (EM, ×8000) demonstrating microvilli. The Figure 2D shows that the microvilli of enterocytes at the base of villi were longer after the HCD compared to after the HFD (*p* = 0.034). This was not the case concerning the length of the microvilli in the top region of the villus. The diameter of the microvilli did not differ depending on the diets (Figure 2E). The ultrastructural analysis also showed that the surface enlargement factor based on the microvilli was significantly higher at the base region after the HCD compared to HFD (Figure 2F, *p* = 0.020). No such difference was present in the top region of microvilli.

### 3.3. Ultrastructural Assessment of Mitochondria in the Enterocyte

The number of mitochondria were counted in relation to the nucleus of the enterocyte (see Figure 3A). The *supranuclear part* at the top of the villi demonstrated a significantly increased number of mitochondria following HCD as compared to HFD (Figure 3B). No other part differed in number of mitochondria in relation to diet.

The size of mitochondria was calculated as the area (µm^2^) occupied in the respective parts of the enterocytes in relation to the number of mitochondria (Figure 4). It was noticed that the supranuclear part in the top region of the villus had mitochondria that were significantly smaller after HCD compared to HFD. The situation was the other way around in the infranuclear part of the base region, where HCD caused increased mitochondrial area as compared to the HFD. The cristae in the mitochondria were more homogenous and the amounts were *increased* in HFD compared to HCD in all but the infranuclear part in the base region of the villi (Figure 5B). The mean area of cytoplasm in the enterocytes, as well as the number of granula in the mitochondria, were on similar levels independent of the diet (Appendix A).

### 3.4. Confocal Microscopic Assessment of the Mitochondrial Phenotype along Villus-Axis

In an attempt to see if the mitochondria changed their mitochondrial phenotype in relation to the two diets, we quantified the intensities of the ketogenic enzyme HMGCS2 per mitochondrial marker TOMM20. After immunofluorescence staining, z-stack images were acquired with confocal scanning microscopy and the intensity of each staining over the stack was quantified for each location (see example in Figure 6A and Appendix A). For this analysis we also included the middle region of the villi, i.e., the part situated between the base and the top region. In both the middle and top regions of the villus, HFD resulted in increased HMGCS2 per TOMM20 (*p* = 0.045 and 0.0074, respectively) compared to the HCD, while in bases of the villi the diet had no effect (Figure 6B).

## 4. Discussion

In the present study, healthy volunteers were served two iso-caloric diets dominated by either carbohydrates or fat, in a randomized crossover design study. The HCD showed increased mitotic cells in the crypts compared to the HFD. This suggests an increased crypt-to-villus migration rate and an increased epithelial turnover rate following the HCD. Few or no signs of increased apoptosis in the biopsies were noted. The morphometric analysis on the villus structure did not demonstrate any change of the epithelial enlargement factor. The primary variable of the study was therefore negative. However, the ultrastructural analysis demonstrated a significant increase in the enlargement factor at the bases of the villi, probably due to elongation of microvilli following the HCD compared to the HFD. Apparently, the HCD induced enlargement of the basal parts of the microvilli which carry the transport capacity, mainly for sugars. This goes hand in hand with our previous report that also showed an increased ability of glucose uptake via the sodium glucose transporter 1 (SGLT1) following HCD compared to HFD [8].

Mitochondria play a crucial role in the lifecycle of the cell by performing energizing oxidative reactions involving amino acids, lipids and ketone bodies, and create the vast majority of adenosine triphosphate (ATP) that is the energy storage necessary to support all cellular functions. The term “mitochondria”, from the Greek μίτος, *mitos*, “thread”, *chondrion*, “granule”, was coined in 1898. The first high-resolution electron micrographs of mitochondria were published in 1952. In 1957, the term “powerhouse of the cell” was coined by Philip Siekevitz [12], and in 1967, ribosomes were found in mitochondria. The enterocytes are elongated with a polar shape with the microvilli pointed towards the lumen, with the nucleus in the middle. Therefore, it is convenient to present the mitochondria in the supranuclear part as well in an the infranuclear part (see Figure 3A). Previous studies have shown that mitochondria control differentiation and that the number of mitochondria per cell doubles as cells migrate from the crypt base to the villus top [13,14,15]. In our experiments, the number of mitochondria increased in the supranuclear part of the top region of the jejunal villi following HCD compared to HFD. The size of mitochondria, however, was smaller in this part of the villus. An increased number of small mitochondria may therefore be a sign of an increased mitochondrial division rate [14]. Furthermore, the size of the mitochondria increased in the infranuclear part of the base region of the villi. The largest mitochondria, of some 5.3 µm, were found in the villi bases infranuclearly after HCD, where the enterocytes are formed and start their development along the villus crypt-axis. In our electron micrographs, most mitochondria appeared as spherical or slightly oval in shape. Some mitochondria were elongated, and some of them were in contact with other, often elongated, mitochondria. Interestingly, not a single occurrence of mitochondrial division was visible throughout our work. It is thought that *matrix granules* may create a level of “connectivity” between the inner and outer mitochondrial membranes that allows for homeostasis of the mitochondrion [16]. We found several granules in the mitochondria of the enterocytes, but there were no differences between the two diets (see Appendix A). However, the number of *cristae* within the mitochondria was significantly lower at the tops (both in the supra- and infranuclear parts), as well as in the supranuclear part in villi subjected to HCD compared to HFD. The infranuclear mitochondria at the bases showed a similar pattern, but this did not attain statistical significance. The present results thus demonstrated an increased number of mitochondrial cristae in the enterocytes upon following the HFD compared to following the HCD. The cristae host the majority of respiratory enzymatic actions and energy production in the form of ATP.

In the mitochondria, the energy-rich ketone bodies can be produced. We have recently shown that the jejunum of the small intestine can express the rate-limiting ketogenic enzyme mitochondrial 3-hydroxy-3-methylglutaryl-CoA synthase (HMGCS2) and produce ketone bodies during prolonged consumption of a diet high in fat [17]. In another of our previous studies, we showed that expression of jejunal HMGCS2 was increased after HFD [8]. To investigate if the amount of HMGCS2 expression in the enterocytes is linked to the number of mitochondria, we performed confocal scanning microscopy along the villus-axis in a number of experiments. The TOMM20 antibody was used as a marker of the mitochondria in the enterocytes. The intensity ratio between HMGCS2 and TOMM20 revealed that HFD, compared to HCD, increased HMGCS2 with the largest values at the tops of the villi. These results thus suggest that the increased number of mitochondria is linked to an increased expression of HMGCS2. The presence of mitochondrial HMGCS2 seems to be involved in reducing SGLT1 expression and is regulated by the dietary fat [8]. Thus, our morphological study showed that the jejunum is metabolically very flexible and induces a metabolic reprogramming of these cells dependent on the composition of the diet and the substrate load, fat or carbohydrate. However, different types of fat (saturated and unsaturated) may affect the metabolic adaptation [18]. Here we used a mix but with predominantly saturated fat. Regardless, these findings may represent a step forward in understanding the influences of the diet on the intestine, as well as their interactions in the development of obesity.

In summary, a morphometric study was done on the jejunal mucosa in healthy volunteers who consumed both a HCD and a HFD in a random crossover fashion. The study did not demonstrate any change in the mucosal surface enlargement factor, but indicated an ultrastructural larger surface enhancement of the microvilli following the HCD compared to the HFD. The mucosal epithelium also showed signs of increased mitotic activity after HCD compared to HFD, but no clear evidence of increased apoptosis was found. The ultrastructural investigation of the mitochondria showed an increased number of cristae in the mitochondria (indicating a larger metabolic activity) following the HFD compared to the HCD. Confocal microscopic assessment found the highest HMGCS2 count per mitochondrial marker in the tops of the jejunal villi following HFD compared to HCD. The study supports that there is a morphometric adaption in the jejunal mucosa following the two-week diets, which is not mainly on a histological level, but rather on the ultrastructural level.

## Figures and Tables

**Figure 1 nutrients-14-04123-f001:**
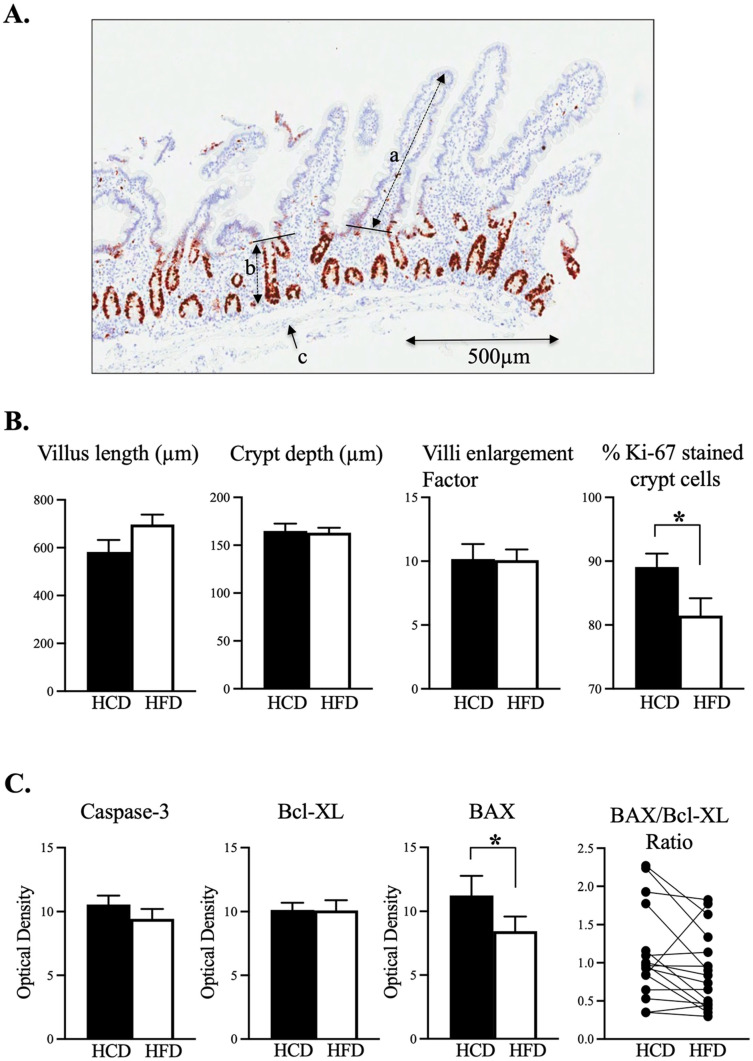
(**A**). The upper panel shows one example of a sectioned biopsy with a haematoxylin background and Ki-67 stained (brown). Villus length (a) and crypt depth (b) were assessed only where tissue orientation was optimal and allowed identification of tip of villus, transition from villus to crypt and the crypt bottom. Villi enlargement factor was assessed using the gridline intersection method of the surface epithelium in relation to muscularis mucosae (the latter depicted in c). (**B**). Villus length, crypt depth and villi enlargement factor were on a similar magnitude following HCD or HFD. Ki-67 cells in the crypts were significantly more stained in relation to all cells after HCD compared to after HFD (*p* = 0.027). (**C**). The lower panel shows the apoptosis biomarkers Caspase-3 and Bcl-XL, which did not differ between diets. BAX was increased following HCD as compared to HFD (*p* = 0.034). However, the ratios of BAX/Bcl-XL were on a similar level independent of diet. Data are means + SEM; n = 11–15; * *p* ≤ 0.05 Wilcoxon signed-rank test.

**Figure 2 nutrients-14-04123-f002:**
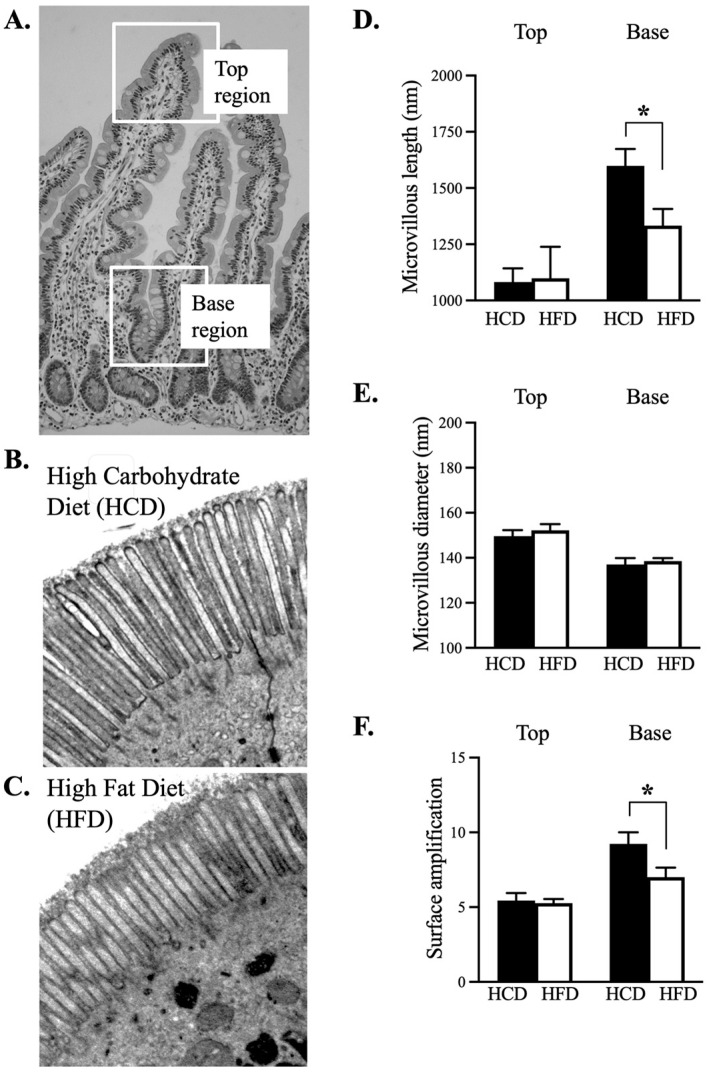
Panel (**A**) The study focused on the enterocytes with a position at the *top* region and at the *base* region of the villus (light microscopy: ×10). Panels (**B**,**C**) Example of TEM-based micrographs (×8000) from the same individual after 2 weeks HCD and after 2 weeks HFD. Panels (**D**,**F**) Shows that microvillous was significantly longer following the HCD (*p* = 0.034) compared to HFD, and that the enlargement factor was also significantly increased following HCD (*p* = 0.020) compared to the HFD. Panel (**E**) The microvillous diameter was on a similar level independent of ingestion of the HCD or HFD. Please note that the length of the microvilli is generally longer while the diameter is thinner at the base region. Data from the whole examined group (n = 6) are depicted as means + SEM of pairwise comparisons, * *p* ≤ 0.05 Wilcoxon signed-rank test.

**Figure 3 nutrients-14-04123-f003:**
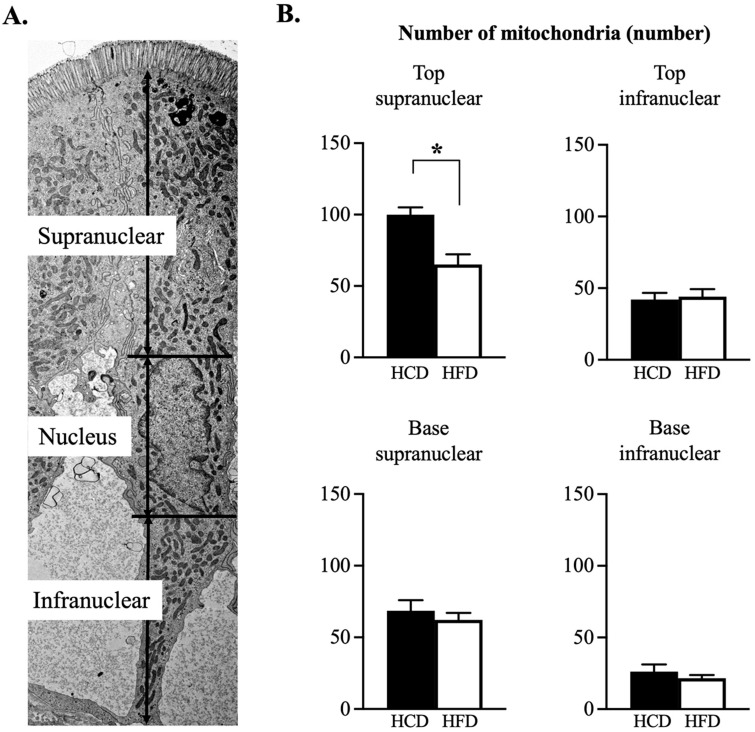
(**A**). The figure shows one enterocyte where the *supranuclear* and *infranuclear* parts are depicted in relation to the *nucleus*. (**B**). The number of mitochondria was found to be increased in the supranuclear part of top region during HCD (*p* = 0.012) compared to HFD. No differences were noted in the top region infranuclear part, as well in the base regions. Electron microscopy: 4 image ×4000, n = 6 pairwise comparisons, * *p* ≤ 0.05 Wilcoxon signed-rank test.

**Figure 4 nutrients-14-04123-f004:**
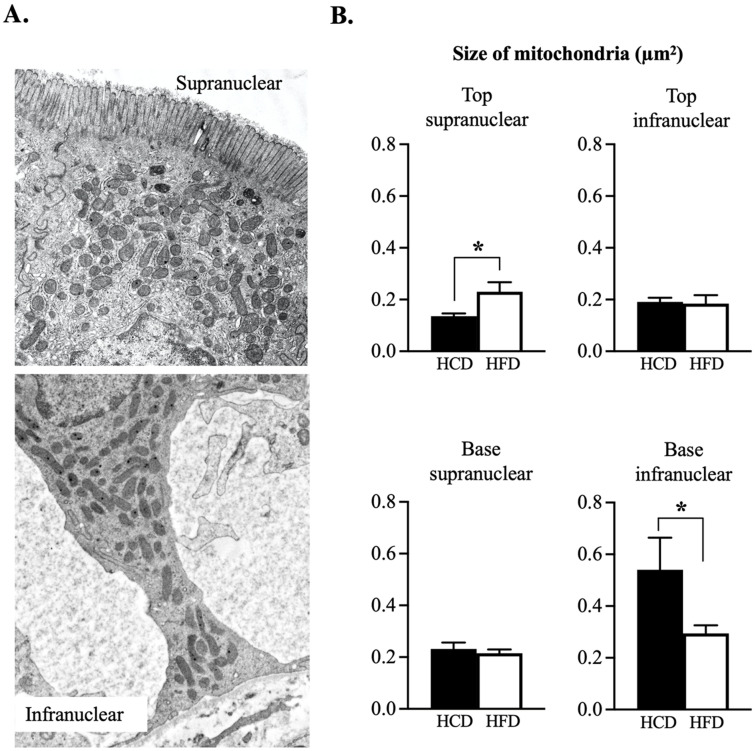
(**A**). The figure shows examples of mitochondria in the supranuclear and infranuclear parts where most mitochondria appeared to be spherical or slightly oval in shape. (**B**). The *size* of mitochondria showing that the top supranuclear had increased following HFD (*p* = 0.02) compared to HCD. The opposite was the case at the base, infranuclear part, where the size increased with HCD (*p* = 0.02) compared to HFD. No differences were noted in the top region infranuclear part, as well in the base region supranuclear parts of the enterocyte. Electron microscopy: ×8000, n = 6 pairwise comparisons, * *p* ≤ 0.05 Wilcoxon signed-rank test.

**Figure 5 nutrients-14-04123-f005:**
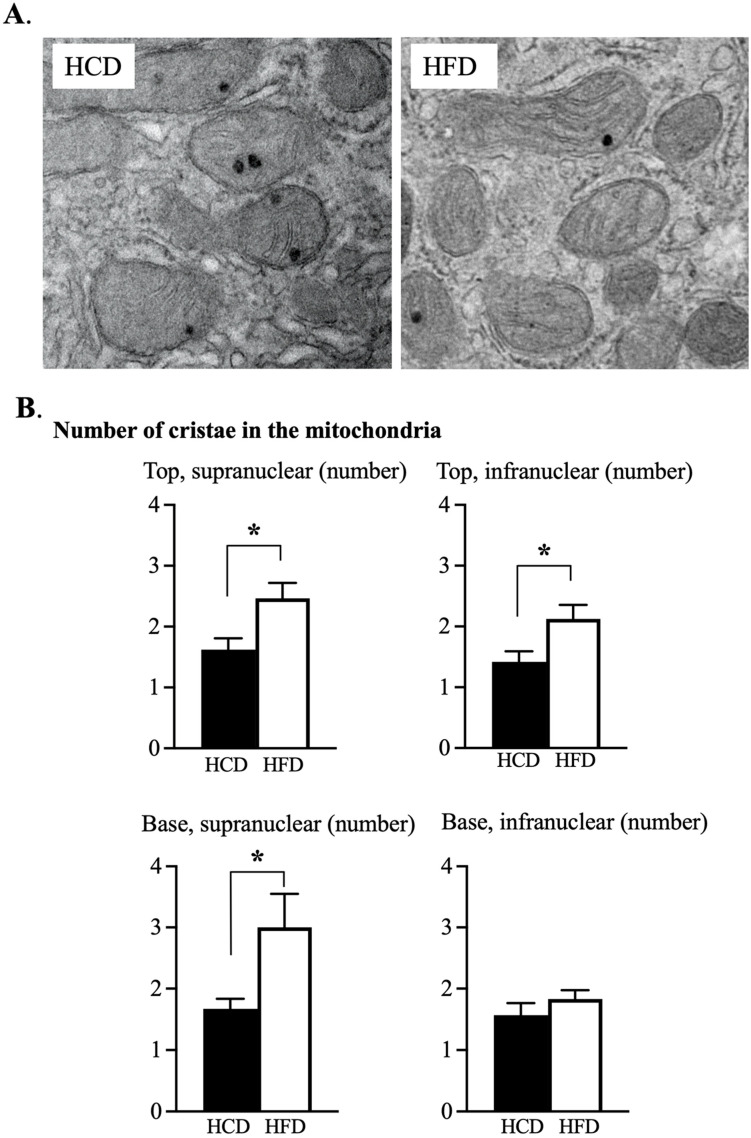
(**A**). Shows two examples of mitochondria from the same participant at a magnification ×31,500 after ingestion of HCD and HFD over 2 weeks. (**B**). The number of cristae in the mitochondria: Note that ingestion of the HFD compared to HCD increased the number of cristae in the top region (*p* = 0.01 and *p* = 0.02 respective), as well in the base supranuclear part (*p* = 0.03) of the enterocyte. Electron microscopy: ×31,500, n = 6, * *p* ≤ 0.05 Wilcoxon signed-rank test.

**Figure 6 nutrients-14-04123-f006:**
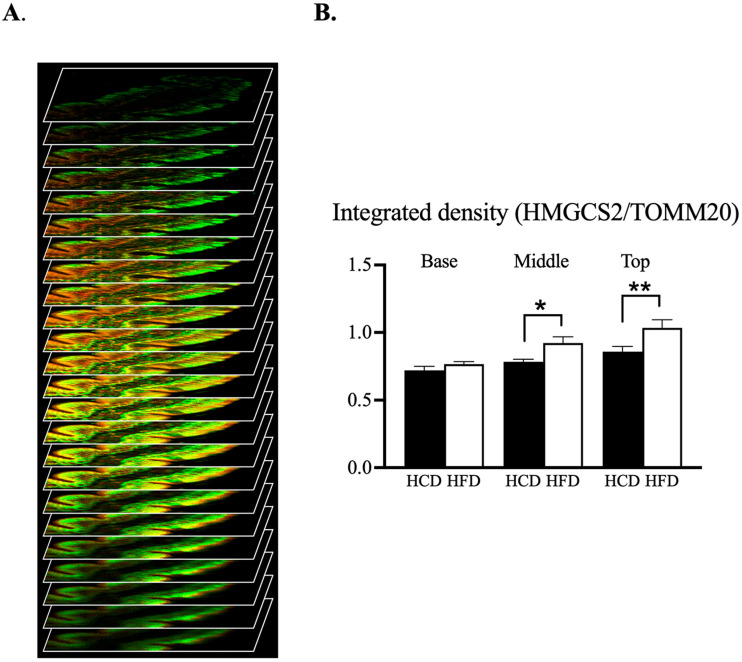
(**A**). Graphical representation of optical sectioning from confocal scanning microscopy of villus. (**B**). Confocal microscopy showing intensity of HMGCS2 per TOMM20 at different locations (base, middle and top) of jejunal villus after 2 weeks of HCD and HFD. Data are given as mean + SEM; n = 9 (number of individuals = 3), 2-way ANOVA with Sidak’s multiple comparison test used for statistical analysis, * *p* ≤ 0.05, ** *p* ≤ 0.01.

## Data Availability

All authors had access to the study data and reviewed and approved the final manuscript. All data relevant to the study are included in the article or uploaded as Appendix A. Individual participant data will not be shared.

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
