# Peer review of "Morphological Adaptation in the Jejunal Mucosa after Iso-Caloric High-Fat versus High-Carbohydrate Diets in Healthy Volunteers: Data from a Randomized Crossover Study"

_nutrients, 2022, doi:10.3390/nu14194123_

Round 1
Reviewer 1 Report
In the presented manuscript the authors aimed to the conditions for jejunal glucose absorption in healthy man. The study has potential, but there are a few points that must be improved.
1. In introduction section, the novelty of the present study should be more specific.
2. Please highlight the physiological significance of the morphological adaptation of the jejunal mucosa following an iso-caloric high-fat and high-carbohydrate diet in the Discussion section.
3. The abbreviations of HCD and HFD for high-carbohydrate diet and high-fat diets are shown in the abstract, but there are still multiple sentences using “high-carbohydrate diet (HCD) or high-fat diet (HFD)” in the paper.
4. In line 72, it would be better to replace “the first report” with “a first a report”.
5. There are two “but” in the last sentence of the manuscript, please double check the sentence.
Author Response
AUTHORS RESPONSE TO REVIEWER #1
Reviewer 1.
In the presented manuscript the authors aimed to the conditions for jejunal glucose absorption in healthy man. The study has potential, but there are a few points that must be improved.
- In introduction section, the novelty of the present studyshould be more specific.
AUTHORS:
1.We thank the Reviewer for giving valuable criticism and advice. We have now clarified the purpose and originality of the study in the introduction (page 3) and also added a new reference to clarify the area of ​​interest.
Reviewer 1.
- Please highlight the physiological significance of the morphological adaptation of the jejunal mucosa following an iso-caloric high-fat and high-carbohydrate diet in the Discussion section.
AUTHORS:
We have now added a paragraph at the end of the discussion to relate more to the purpose of the study and why it has physiological significance.
Reviewer 1.
- The abbreviations of HCD and HFD for high-carbohydrate diet and high-fat diets are shown in the abstract, but there are still multiple sentences using “high-carbohydrate diet (HCD) or high-fat diet (HFD)” in the paper.
AUTHORS:
3.We thank the reviewer for your attention. We have now changed so that we use the abbreviations throughout the text.
Reviewer 1.
- In line 72, it would be better to replace “the first report” with “a first a report”.
AUTHORS:
- We have now changed the sentence to “a first report”.
Reviewer 1.
- There are two “but” in the last sentence of the manuscript, please double check the sentence.
AUTHORS:
5.We thank the reviewer for your attention. We have now changed the sentence.
Reviewer 2 Report
Casselbrant et al have found that 2 weeks of diet intervention can induce mitochondrial changes in specific regions of the jejunum of adult humans. This study and manuscript are of high quality and it was a pleasure to read.
One recommendation is to provide the exact foods from the diets in the methods section. Specifically the high fat diet. Was it high in saturated fatty acids or unsaturated fatty acids? These different types of lipids are known to exert opposing effects with regards to apoptosis and mitochondrial metabolism of energy substrates.
Additionally, an extra paragraph describing the findings in relation to the type of fats consumed would also be a great discussion angle for the manuscript.
Author Response
AUTHORS RESPONSE TO REVIEWER #2
Reviewer 2.
Casselbrant et al have found that 2 weeks of diet intervention can induce mitochondrial changes in specific regions of the jejunum of adult humans. This study and manuscript are of high quality and it was a pleasure to read.
- One recommendation is to provide the exact foods from the diets in the methods section. Specifically the high fat diet. Was it high in saturated fatty acids or unsaturated fatty acids? These different types of lipids are known to exert opposing effects with regards to apoptosis and mitochondrial metabolism of energy substrates.
AUTHORS:
1.We thank the Reviewer for giving valuable criticism and advice. We have now included a specific reference to the exact food our participants ate in the methods section (available online at https://www.mdpi.com/article/10 .3390/nu13103322/s1). We have also clarified in the text the contents of the food further (page 5). The food contained both saturated, monounsaturated as well as polyunsaturated fat, but with a higher percentage of saturated fat.
Reviewer 2.
- Additionally, an extra paragraph describing the findings in relation to the type of fats consumed would also be a great discussion angle for the manuscript.
AUTHORS:
- We thank the reviewer for your imput about this. We have added a paragraph to the discussion (page 16) as well as a reference that highlights the importance of different fats on mitochondrial metabolism. However, we cannot draw any conclusions about our results depending on the fat content of the diet.